# Genome analysis of the monoclonal marbled crayfish reveals genetic separation over a short evolutionary timescale

Olena Maiakovska[1], Ranja Andriantsoa[1], Sina Tönges[1], Carine Legrand[1], Julian Gutekunst[1], Katharina Hanna[1], Lucian Pârvulescu [2], Roman Novitsky[3], András Weiperth[4], Arnold Sciberras[5], Alan Deidun[5], Fabio Ercoli [6,7], Antonin Kouba[8] & Frank Lyko [1✉]

The marbled crayfish (*Procambarus virginalis*) represents a very recently evolved parthenogenetic freshwater crayfish species that has invaded diverse habitats in Europe and in Madagascar. However, population genetic analyses have been hindered by the homogeneous genetic structure of the population and the lack of suitable tools for data analysis. We have used whole-genome sequencing to characterize reference specimens from various known wild populations. In parallel, we established a whole-genome sequencing data analysis pipeline for the population genetic analysis of nearly monoclonal genomes. Our results provide evidence for systematic genetic differences between geographically separated populations and illustrate the emerging differentiation of the marbled crayfish genome. We also used mark-recapture population size estimation in combination with genetic data to model the growth pattern of marbled crayfish populations. Our findings uncover evolutionary dynamics in the marbled crayfish genome over a very short evolutionary timescale and identify the rapid growth of marbled crayfish populations as an important factor for ecological monitoring.

[1] Division of Epigenetics, DKFZ-ZMBH Alliance, German Cancer Research Center, Im Neuenheimer Feld 580, 69120 Heidelberg, Germany. [2] Department of Biology-Chemistry, West University of Timisoara, 16A Pestalozzi St., 300115 Timisoara, Romania. [3] Department of Water Bioresources and Aquaculture, Dnipro State Agrarian and Economic Universities, 25 Serhii Efremov St., Dnipro 49600, Ukraine. [4] Department of Aquaculture, Institute for Natural Resources Conservation, Faculty of Agriculture and Environmental Sciences, Szent István University, Páter Károly utca 1, 2100 Gödöllő, Hungary. [5] Physical Oceanography Research Group, Department of Geosciences, University of Malta, Msida MSD 2080, Malta. [6] Chair of Hydrobiology and Fisheries, Institute of Agricultural and Environment Sciences, Estonian University of Life Sciences, Kreutzwaldi 5D, 51006 Tartu, Estonia. [7] Natural Resources and Environment, Department of Biological and Environmental Science, University of Jyväskylä, P.O. Box 35, 40014 Jyväskylä, Finland. [8] Faculty of Fisheries and Protection of Waters, CENAKVA, University of South Bohemia in České Budějovice, Zátiší 728/II, 38925 Vodňany, Czech Republic. ✉email: f.lyko@dkfz.de

Parthenogenetic reproduction, i.e., the development of embryos from unfertilized eggs, has been described in many groups of animals[1]. Arthropod examples include parthenogenetic lineages of silkworms (*Bombyx mori*), water fleas (*Daphnia pulex*), weevils (*Otiorhynchus scaber*), and stick insects (*Timema* spp.)[2–5]. Complete genome sequences from parthenogenetic animals have presented unique features, and it has been suggested that parthenogenetic genomes evolve toward effective haploidy, as predicted by the Meselson effect[6,7]. However, these processes require parthenogenesis over millions of years. Also, a comparative analysis of 24 parthenogenetic genomes revealed a substantial degree of diversity in their features, which reflects the considerable biological differences among the corresponding species[8].

Marbled crayfish (*Procambarus virginalis*) are obligate parthenogenetic freshwater crayfish that were first described in the German aquarium trade[9,10]. The species is a recent descendant of the slough crayfish (*Procambarus fallax*), which is an abundant, sexually reproducing freshwater crayfish species in peninsular Florida and southern Georgia[11]. Marbled crayfish may have originated from *P. fallax* through an evolutionary very recent macromutation, which resulted in instant reproductive isolation[12]. All known animals can likely be traced back to an initial population that was founded in 1995[10]. As such, the evolutionary age of the species appears to be ~25 years.

Parthenogenesis can cover a variety of reproductive mechanisms[13]. Automixis is characterized by the fusion of meiotic products, with the production of genetically distinct offspring due to genetic recombination. Apomixis, on the other hand, is characterized by a bypass of meiotic recombination, resulting in clonal progeny that represents an exact copy of the maternal genotype. Early reproductive studies, in combination with microsatellite analyses, suggested that marbled crayfish utilize apomictic parthenogenesis, which results in genetically uniform offspring[14,15]. The absence of meiosis during oocyte maturation was also concluded from histological studies of oocyte nuclei[16]. More recently, it has been described that the first meiotic division is defective in marbled crayfish, but the chromosomes recover and proceed through the second meiotic division[17]. While this finding is consistent with apomictic parthenogenesis, the absence of meiotic recombination events in marbled crayfish remains to be confirmed.

A de novo draft assembly of the marbled crayfish genome with a genome size of approximately 3.5 Gbp and >21,000 genes has been published recently[18]. It is characterized by a triploid AAB genotype, with a duplicated "A" haplotype resulting from autopolyploidization and a heterozygous "B'" haplotype presumably resulting from fertilization[18]. Comparative whole-genome sequencing (WGS) of animals from various aquarium lineages and from wild populations suggested that the marbled crayfish meta-population represents a single, genetically homogeneous clone[18]. Genetic variants are generated by the ongoing accumulation of natural mutations[18], but their numbers are greatly limited by the extremely young evolutionary age of the species.

Due to their overall robustness and aesthetic appeal, marbled crayfish have become popular and widely distributed aquarium pets[19]. In parallel, the animals have increasingly colonized wild habitats. It is generally assumed that these populations were founded by anthropogenic releases and thus represent offspring of a single animal. This is exemplified by the situation in Madagascar, where the animals were first detected around 2006[20,21], and have now populated an area of 100,000 km$^2$ that encompasses many diverse habitats[18,22]. Isolated wild populations have also been described in Germany and in several other European countries[23–29], but their genomic variation has not been analyzed yet.

The clonality of marbled crayfish genomes presents unique challenges and requirements for the analysis of genomic variation. Marbled crayfish populations are groups of individuals that coexist spatially and temporally, but appear to miss classical attributes of panmictic populations, like random mating of individuals and active recombination. Hence, the population genetic structure of clonal marbled crayfish cannot be characterized by allele frequency dynamics, as the underlying processes are typical for sexually reproducing populations[30,31]. Our study addresses these issues and provides information about the emerging genetic differentiation within the marbled crayfish population.

## Results

**Survey of wild marbled crayfish populations in Europe**. For a systematic genetic comparison of marbled crayfish populations in Europe, we surveyed known populations. This revealed that some reports had described transient appearances of single animals or groups of animals (probably from recent anthropogenic releases), while others described wild populations that have now stably persisted over several years. We restricted our study to stable wild populations and confirmed the taxonomic identity of the animals by genetic authentication of multiple specimens from 15 populations and from nine different European countries (Supplementary Table S1). Our results thus provide a systematic survey of stable wild marbled crayfish populations in Europe (Fig. 1).

**WGS of wild marbled crayfish populations**. While marbled crayfish can be considered as a single genetic clone, they also show a limited number of genetic polymorphisms in their nuclear genomes that result from the accumulation of genetic mutations within lineages over time[18]. Population-specific patterns of polymorphisms allow the analysis of the genetic relationships between populations. We, therefore, obtained WGS data for ten animals, representing ten populations in eight countries (Supplementary Table S2), as well as for four additional animals from a single wild population. The average mapping coverage was >30x for most samples (Supplementary Table S2), thus permitting detailed genetic analyses.

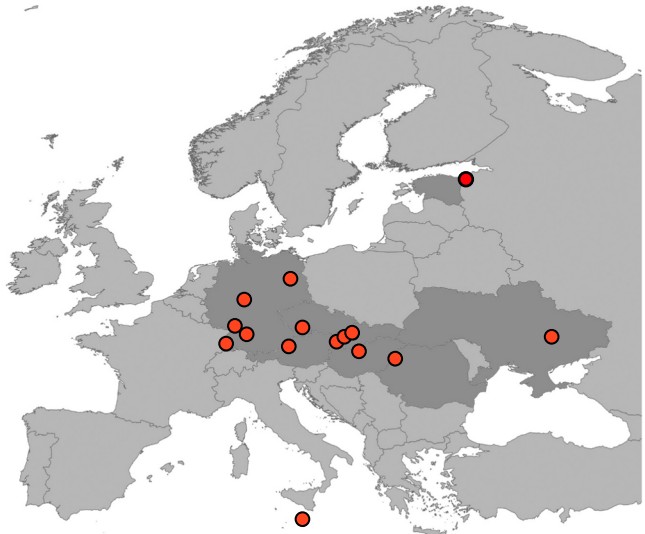

**Fig. 1 Map of Europe, highlighting stable wild marbled crayfish populations, as described in this study.** The map was generated with QGIS 3.2.0, locations of specific populations are indicated by red dots (see Supplementary Table S1 for details).

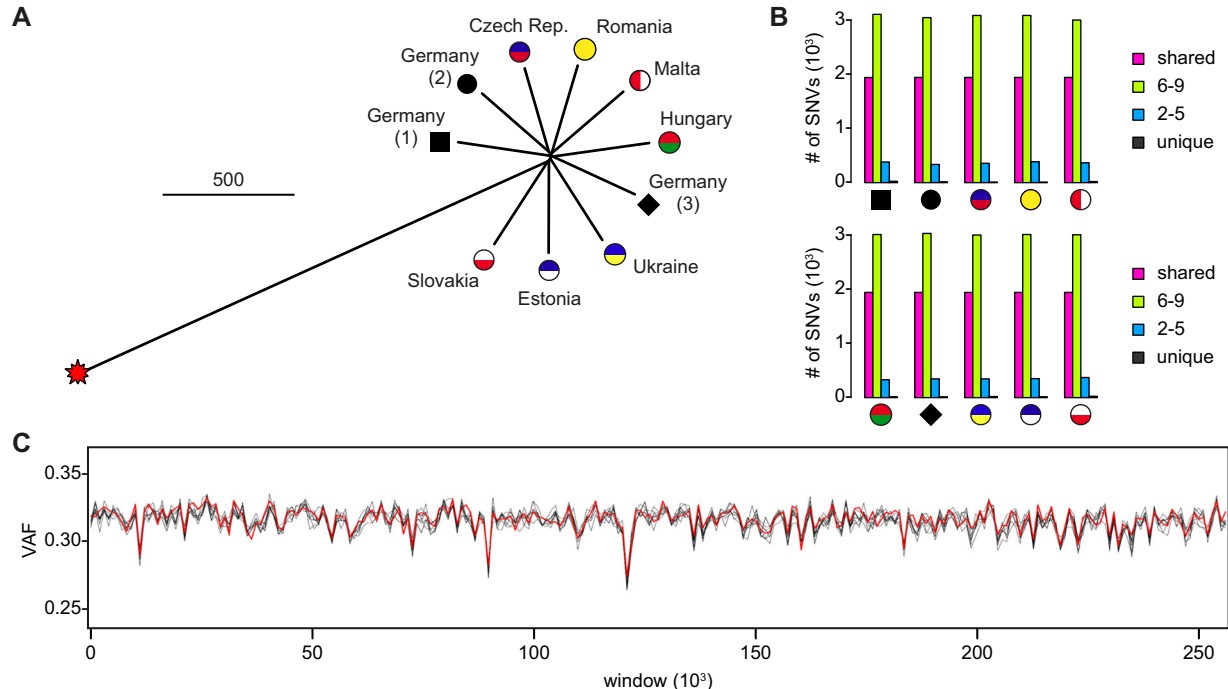

**Fig. 2 Genetic relationships between representative specimens from ten different European populations, as determined by whole-genome sequencing (WGS). A** Distance tree. Scale bar: 500 counts, representing 500 SNV from the set of common SNVs for all samples. The oldest known and likely foundational aquarium lineage of marbled crayfish is included as a reference (red star). **B** Bar plots illustrating the number of unique and shared SNVs. 6–9 and 2–5 indicate the number of SNVs that are shared by 6 to 9 and 2–5 animals, respectively. **C** Genome-wide analysis of variant allele frequency (VAF). Lines represent the average VAF per window of 10,000 heterozygous positions for each of the ten reference specimens. Variant allele frequencies are highly similar for all specimens and do not provide any evidence for loss of heterozygosity. The red line indicates the reference genome.

**Comparative genome analysis**. Comparative analysis of marbled crayfish genomes was based on the identification and pairwise comparison of single-nucleotide variant (SNV) patterns (see "Methods" for details), using the genome sequence of a specimen from the oldest known aquarium lineage of marbled crayfish (founded in 1995) as a reference. The numbers of SNVs ranged from 5722 to 6007. This is an order of magnitude below the frequency of SNVs in clonal daphniids[32] and likely reflects the young evolutionary age of marbled crayfish. A corresponding distance tree showed high genetic similarity among the wild European populations of marbled crayfish and their more distant relationship to the ancestral aquarium lineage (Fig. 2A). Parallel analysis of mitochondrial genomes showed identical sequences for all marbled crayfish specimens and a clear separation from the reference mitochondrial genome of the sexually reproducing mother species, *P. fallax*. (Supplementary Fig. S1). These findings confirm the monoclonality of marbled crayfish from established wild populations across Europe and also suggest emerging genetic differentiation within the meta-population.

Further analysis showed that the proportion of unique, partially shared, and fixed polymorphisms were similar for all representative individuals (Fig. 2B). In addition, the number of unique SNVs was consistently low (Fig. 2B), suggesting minimal numbers of specific mutations for each population. Finally, we also addressed the possibility of low levels of recombination in marbled crayfish. Genome-wide analysis of variant allele frequency (VAF) failed to reveal any evidence for loss of heterozygosity (Fig. 2C). These results are consistent with the notion that marbled crayfish reproduce by apomictic parthenogenesis[14,15] and suggest that genetic variation in marbled crayfish is driven by natural mutation.

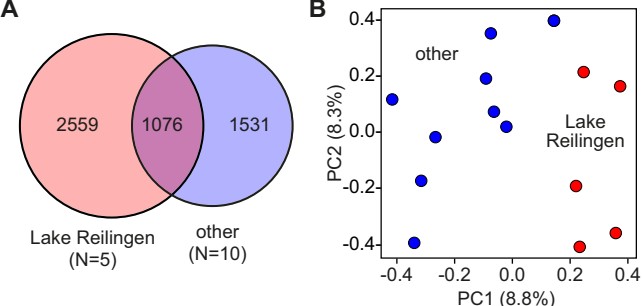

**Fig. 3 Analysis of multiple specimens from Lake Reilingen, in comparison to representative specimens from other European populations. A** Venn diagram demonstrating the level of overlap of shared SNVs between the Lake Reilingen and other European populations. From the complete set of 7858 polymorphic sites, 3635 were shared among the Lake Reilingen specimens, and 2607 were shared among the other European specimens. **B** Principal component analysis (PCA) of multiple specimens from Lake Reilingen (red dots) and other European populations (blue dots).

**Incipient genetic separation of marbled crayfish populations**. To characterize the genomic variation of a single population, we analyzed WGS datasets from five animals from the type locality (Lake Reilingen, Germany). After processing the dataset through the previously established pipeline (Supplementary Table S2), a set of 3635 shared SNVs was identified in the five genomes, which showed a distinct overlap with the set of 2607 shared SNVs in the nine genomes from other European populations (Fig. 3A). Principal component analysis (PCA) on the complete set of SNVs from these specimens showed that the five samples from Lake Reilingen were clearly separated from the other samples (Fig. 3B),

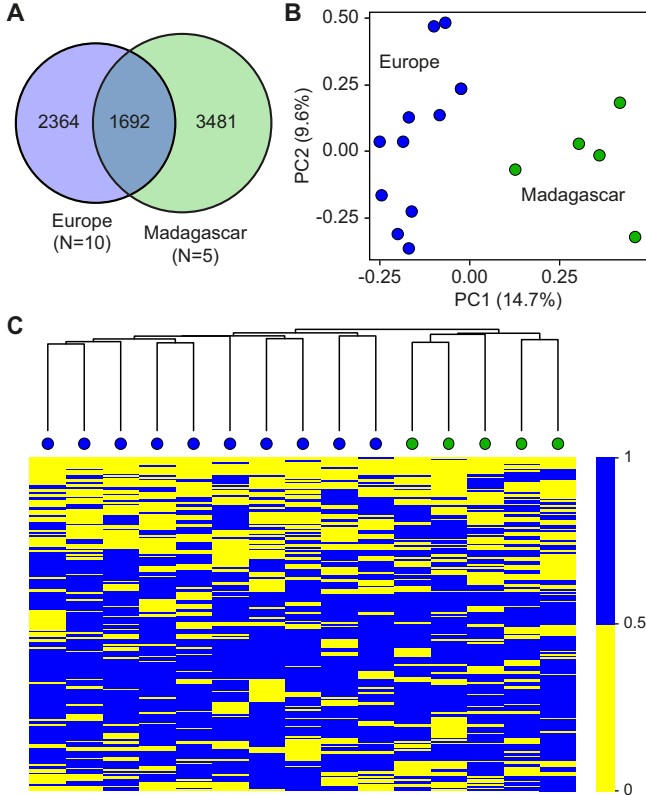

**Fig. 4 Comparison of European and Malagasy populations, based on common genetic variants. A** Venn diagram demonstrating the level of overlap of shared single-nucleotide variants (SNVs) between the European and Malagasy populations. **B** Principal component analysis (PCA) of representative specimens from ten European and five Malagasy populations. **C** Heatmap of SNV patterns, based on the 8912 common polymorphic sites. The color represents the average number of SNVs per site, as indicated. The clustering reflects the specific geographical SNV patterns.

only 4% were located in coding regions (Supplementary Table S3). Moreover, the analysis of base changes showed a higher abundance of G to A and C to T transitions (Supplementary Table S4). Similar patterns have also been observed in other parthenogenetic arthropods[33,34].

**Size estimate and growth simulation of a marbled crayfish population.** It is known that marbled crayfish populations can grow rapidly[18], but population size estimates are currently not available. We, therefore, sought to apply our genetic analysis towards a better understanding of marbled crayfish population growth. Bathymetric mapping of Lake Reilingen (Fig. 5A) was used for planning a mark-recapture survey[35] and revealed a maximum depth of 17 meters (Fig. 5B). In total, 18 traps were deployed (Fig. 5C), which were baited and checked daily for ten consecutive days. During this time period, a total of 394 animals was collected (Supplementary Table S5). 96% of these animals exceeded the trapping size cutoff of >6 cm (Supplementary Fig. S2), which was determined by the mesh width of the traps. After data processing (Supplementary Fig. S3), Schnabel's formula was applied, resulting in a population size estimate of 22,962 (SE = 3613) sexually mature animals >6 cm. Further sampling and data analysis (see "Methods" for details) resulted in a total population size estimate of 192,000 (SE = 67,000) animals. The high error of the latter estimate is largely due to the difficulty to accurately determine the number of small animals in the lake. Nevertheless, the population size and corresponding population density is consistent with data reported for the closely related freshwater crayfish species *P. fallax* in Florida[36].

Finally, we used our WGS datasets from Lake Reilingen to make inferences about the growth model of the population. In a first step, the SNV dataset of the four sequenced specimens from the population was used to estimate the total genetic variation occurring within the population ($N = 192,000$ animals). Subsequently, we applied three different growth models (logistic growth, exponential growth, and growth based on the Allee effect) to simulate the population growth, based on a range of physiological parameters for marbled crayfish reproduction (see "Methods" for details). We then calculated the probabilistic total genetic variability for the populations with different simulated growth models considering a well-established arthropod reference mutation rate[37]. Further analysis showed that the genetic variability with a simulated exponential growth intersects with the variability of the current Lake Reilingen population after 3.7 (±0.3) years (Fig. 5D). These findings suggest that marbled crayfish populations can have a very rapid initial growth phase until they reach saturation due to habitat resource limitations.

## Discussion

The analysis of asexual genomes can inform on important evolutionary aspects of asexual reproduction[8]. Our study provides a genetic analysis of the marbled crayfish, an evolutionary very young species with a monoclonal population structure[10,18]. We confirm overall monoclonality and the absence of recombination in the marbled crayfish genome, consistent with the notion that marbled crayfish reproduce by apomictic parthenogenesis[14,15]. We also uncover the emergence of geographically stratified genotypes and provide inferences about the rapid growth of local colonies.

Clonal genomes present unique challenges for their analysis. We provide an approach for the analysis of clonal population patterns in the absence of allele frequency dynamics by using WGS for the detection of SNVs. In the context of the low genomic diversity within the populations, we applied PCA to

which is consistent with the close genetic relationship of animals from specific populations.

In subsequent steps, we integrated the European WGS datasets with previously published WGS datasets from Madagascar[18] and a newly generated dataset from Madagascar (Supplementary Table S2). After data processing and variant calling for the common dataset, we defined a set of 7537 SNVs from the European ($N = 10$) and Malagasy ($N = 5$) samples (Fig. 4A). These analyses indicated a lower number of shared polymorphisms for the European population (Fig. 4A), which is likely influenced by the higher number of European samples in the analysis. Furthermore, the considerable number of shared polymorphisms between European and Malagasy animals (Fig. 4A) likely reflects the distance of the wild populations from the ancestral aquarium lineage, which provided the reference genome sequence[18]. Furthermore, PCA showed clear segregation into two clusters according to the geographic region of origin (Fig. 4B). The visualization of SNV patterns also suggested the presence of defined clusters with population-specific variation patterns for the European and Malagasy populations (Fig. 4C), consistent with the existence of population-specific genetic signatures. Together, these findings illustrate the emerging differentiation of the marbled crayfish genome in geographically separated populations.

Finally, we also annotated the complete set of SNVs ($N = 16,564$) that was identified in our analyses. The results showed that the majority (74%) were located in intergenic regions and

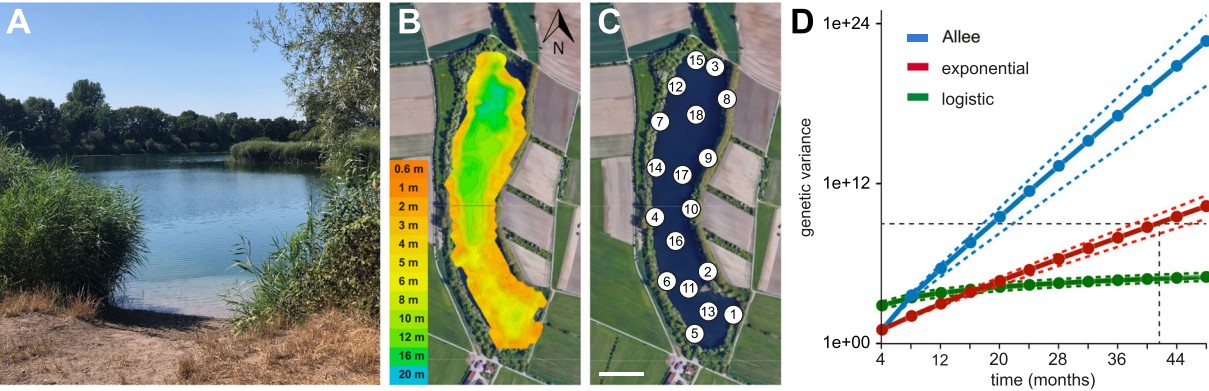

**Fig. 5 Marbled crayfish population size estimate and growth simulation at Lake Reilingen. A** Picture of Lake Reilingen, a small (9 ha) lake in Germany with an established marbled crayfish population. Picture provided by Sina Tönges. **B** Bathymetric map of Lake Reilingen, generated with Deeper Smart Sonar software. Water depths are indicated by color, North is indicated by the arrowhead. **C** Positions of the 18 traps used for the population size estimate. Scale bar: 100 m. Map data: Google, Maxar Technologies. **D** Marbled crayfish population growth kinetics simulations. The horizontal dashed line represents the genetic variability of 1e+09 in the current Lake Reilingen population. Colored lines show the predicted genetic variability for exponential, logistic, and Allee growth models (see "Methods" for details). Dashed colored lines indicate error margins (SE).

detect the emerging divergence. The small number of population-specific mutations is consistent with the young age (<20 years) of the European and Malagasy populations, respectively[20,21,23].

The lack of sexual reproduction and genetic recombination limits genetic variation and is often considered to constrain phenotypic diversity and environmental adaptation. However, the general-purpose genotype model suggests that asexuality can be evolutionarily successful through the selection of generalist lineages with large phenotypic plasticity that can occupy a variety of ecological niches[38–40]. Key examples are provided by bdelloid rotifers and darwinulid ostracods, which represent ancient generalist asexual organisms with wider geographical distribution and broader ecological tolerance than their sexual parent species. Clonal variation has been demonstrated in darwinulid ostracods and has been attributed to mutation accumulation over long evolutionary timeframes[41,42]. The analysis of marbled crayfish genomes will facilitate our understanding of the mechanisms of asexual genome diversification and adaptation. This could conceivably include the preservation of high heterozygosity levels, the selection of specific genetic variants, and changes in genome regulatory mechanisms[43,44].

In conclusion, our study describes 15 stable populations of marbled crayfish from various European countries. It is very likely that a large number of additional populations has not been identified yet. This can be explained by the recent emergence of the animals, their nocturnal activity and the lack of awareness about their existence. eDNA-based methods[45] might facilitate the detection of additional populations in the future and thus uncover the full extent of the marbled crayfish spread. Of note, our study also revealed a considerable number of animals in a relatively small habitat. The rapid growth of marbled crayfish populations likely represents a key factor in the emerging economic development of the animals in Madagascar[22,46]. Importantly, this rapid growth also suggests that marbled crayfish populations should be monitored to determine their impact on freshwater ecosystems.

## Methods

**DNA extraction and WGS.** Animals were collected from various European wild populations (Supplementary Table S1) and from previously described populations in Madagascar[18,22]. Genomic DNA was isolated and purified from abdominal muscular tissue using a Tissue Ruptor (Qiagen), followed by proteinase K digestion and isopropanol precipitation. The quality of isolated genomic DNA was assessed via agarose gel electrophoresis and/or via TapeStation (Agilent). PCR genotyping was performed using the mitochondrial cytochrome B and the nuclear Dnmt1

markers[18]. Whole-genome sequencing was performed on an Illumina HiSeq platform (DKFZ Genomics and Proteomics Core Facility).

**Read preprocessing.** After quality control using FastQC, reads were trimmed with trimmomatic v.0.32[47]. Low-quality bases and adapters were removed with a leading and trailing Phred score <3 and reads were quality filtered by using a base sliding window of size 4 and an average quality per base below 20. Finally, trimmed reads shorter than 40 bases were removed.

**Mapping and duplicate removal.** Trimmed reads were aligned to the marbled crayfish genome (assembly version 0.4)[18] using Bowtie 2 (version 2.2.6) with default parameters[48]. The resulting alignments were processed and filtered for duplicates using samtools (version 1.3)[49]. Also, alignments on short contigs (≤10 kb) were discarded to avoid loss of time in processing fragmented sequences. Sequencing statistics for each individual are summarized in Supplementary Table S2.

**SNV calling and polymorphic site calculation.** The Bayesian genetic variant detector Freebayes was used for variant calling with parameters set for triploidy, minimum mapping quality of 30, and minimum base quality of 20, to exclude low-quality reads and bases from the analysis. Filtering was based on a minimum read depth of 20, a maximum read depth of 200, and a minimum Phred-scaled quality score of 30 for each sample, using gatk-4.1.3.0[50] and SnpSift[51]. In addition, SNV association with genetic features was analyzed using SnpSift[51]. Filtering for indels was performed by custom filtering pipelines. The comparative analysis included all *P. virginalis* WGS datasets, and a WGS dataset from the oldest known aquarium lineage sample (Heidelberg) as a reference[18]. Only sites that were covered in the reference were included in the analysis. To streamline the analysis in the triploid background, heterozygous sites in the reference were eliminated, which focused the analysis of polymorphic sites on non-heterozygous loci. Polymorphic sites were identified by the presence of a single-nucleotide substitution relative to the reference sequence. Other types of substitutions such as insertions or deletions were also discarded from the analysis.

**Tree construction.** The re-coded matrix was used for the calculation of distances between pairs of individuals followed by neighbor-joining tree estimation with default parameters using the Ape package[52]. Finally, an unrooted phylogenetic tree was generated using the Phytools package[53] and default parameters.

For the mitochondrial tree, WGS reads were trimmed and mapped onto the *P. virginalis* mitochondrial genome reference sequence (GenBank KT074364.1). After filtering for quality parameters and duplicate removal, the alignment files were used for the generation of consensus sequences of mitochondrial genomes. The genetic divergence was calculated by multiple sequence alignment in Clustal Omega with default parameters. The phylogenetic tree was constructed based on maximum likelihood estimation using PhyML[54] with default parameters.

**Loss of heterozygosity analysis.** The initial SNV dataset was used for the selection of heterozygous positions in the reference genome sequence. Sites were filtered for a Phred-scaled quality score >30 and a sequencing coverage >5×. This resulted in 250,453 heterozygous positions for VAF analysis. VAF per site was calculated as the ratio of the alternative allele read count to the total number of reads.

**Population structure analysis**. After variant calling, each genotype of the analyzed sample was permutated with integers from 0 (0/0/0) to 3 (1/1/1) according to the alternative allele dosage. A PCA for each set of markers was run using the R package PCAMETHODS[55]. The heatmap was generated based on the permutated matrix using the Pheatmap R package with default clustering parameters and a row aggregation parameter of kmeans_k = 200.

**Population size estimate**. The population size estimate was performed at Lake Reilingen (49.296893N, 8.544591E) in July 2018. A bathymetric map of Lake Reilingen was established using a Smart Sonar PRO + (Deeper) equipped with a GIS system and sonar. A total of 18 traps (mesh width: 1 cm) were deployed at various water depths (Fig. 5C and Supplementary Table S5), with an average distance of 105 m between traps. Traps were baited with rotten chicken meat and fish food every morning before 10:00. Traps were checked every 24 h, and all newly caught animals were marked with nail polish (each trap with a unique color), and released back at the catch site. This protocol was followed for 10 consecutive days to reduce confounding effects arising through animal reproduction, death, migration, and moulting.

To estimate the population size, Schnabel's method[35] was used (Supplementary Fig. S3A). As the mark-recapture saturation was not reached after 10 days, we used R to run a non-linear least square (NLS) regression to predict the catch size plateau (Supplementary Fig. S3B). More specifically, we used a negative exponential equation with two parameters:

$$y = a(1 - e^{-cx}) \tag{1}$$

where c is replaced by (1/e) to run the model on R (function AR.2 from the R package DRC). Furthermore, animal density per square meter was calculated from three sampling points of the lake through hand-catching with small nets. The size distribution of the hand-catches was also used to extrapolate the total number of animals from the number of sexually mature animals in the traps. Finally, numbers were corrected for the bathymetric profile of the lake: for a depth ≤4 m, covering 47,500 $m^2$ of the lake and ten traps, each trap was assumed to be the sampling point for 4750 $m^2$. For a depth of 5–10 m, covering 16,000 $m^2$ and four traps, each trap was assumed to be the sampling point for 4000 $m^2$. For a depth greater than 10 m, covering 29,000 $m^2$ and four traps, each trap was assumed to be the sampling point for 7250 $m^2$.

We evaluated the standard error (SE) by taking into account the uncertainties of the Schnabel estimate, of parameter $a$ (from the negative exponential above), of the surface covered by each trap (arbitrarily fixed SE = 10%), and the ratios of the considered populations relative to all measured populations (bootstrap, 10,000 replicates).

**Population growth simulation**. To simulate marbled crayfish population growth kinetics, the observed population estimates, a growth rate of 200 (±100) offspring per animal and per year, and a mutation rate of $3.6 \times 10^{-9}$ per basepair and per year[37] were used. Growth models included classical exponential growth, logistic growth, and growth based on a strong Allee effect. The classical exponential growth model describes a population size at any time point $t$ by

$$N(t) = N(t_0)e^{g(t-t_0)} \tag{2}$$

and the population size

$$N(i + 1) = N(i)e^{g(t_{(i+1)} - t_i)} \tag{3}$$

by initializing $N_{(t0)} = 1$ for a clonal population with a growth rate of (exp(g) = 200 ± 100) per year. The logistic population growth model is described by

$$N(i + 1) = N_i + gN_i \left(1 - \frac{N_i}{N_{max}}\right) \tag{4}$$

and the population size at time point ti+1- ti, where the maximum number of individuals was set to $N_{max} = 192,000$ (SE = 34.98%) for the current population. The growth with strong Allee effect described by

$$\Delta N = N_{i+1} - N_i = N_i * g * \left(1 - \frac{A}{N_i}\right) \tag{5}$$

where parameter A was chosen according to an established Allee threshold[56] of −3.1.

For calculating the genetic variability of the Lake Reilingen population, the identified SNVs and the estimated mutation rate[37] of the 3,511,656,756 bp marbled crayfish genome assembly[18] were multiplied by population size estimates.

**Statistics and reproducibility**. This study is based on the analysis of WGS datasets of 19 P. virginalis specimens from 15 independent geographic locations. For one population (Lake Reilingen), five independent specimens were analyzed. Variant calling was performed on SNVs with a read coverage >20×. All statistical analyses and visualizations were performed using R (v3.5.0).

**Reporting summary**. Further information on research design is available in the Nature Research Reporting Summary linked to this article.

## Data availability

All sequencing data have been deposited as a NCBI BioProject (accession number PRJNA599283).

## Code availability

Custom computer code is available on Github (https://github.com/OlenaMaiakovska/Population_Analysis_MC.git)[57].

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

## Acknowledgements

We thank Esther Sciberras, Jeffrey Sciberras, Justin Formosa, and Daniela Latzer for sample collection. We also thank many colleagues and the Angelsportverein 1960 Reilingen e.V. for their support of the mark-recapture analysis at Lake Reilingen. F.E. acknowledges funding from the Estonian Ministry of Education and Research (institutional research funding project IUT 21-2 to Tiina Nõges) and from the Estonian Research Council (Mobilitas Pluss research project MOBJD29). This work was supported by funding from the Czech Science Foundation (project no. 19-04431S to A.K.).

## Author contributions

O.M. performed the data analysis and contributed to the paper writing; R.A. and S.T. designed and carried out the mark-recapture analysis; C.L. designed the population growth model; J.G. designed the SNV analysis; K.H. carried out genotyping experiments; L.P., R.N., A.W., A.S., A.D., F.E., and A.K. provided samples; F.L. conceived the study, and wrote the paper with contributions from other co-authors.

## Funding

## Competing interests

The authors declare no competing interests.
