## [Peer Review File · Communications Biology]

Reviewers' Comments:

Reviewer #1:

Remarks to the Author:

In their manuscript entitled "Population genomic analysis of the monoclonal marbled crayfish (*Procambarus virginalis*)" Maiakovska et al. combine Illumina sequencing, bioinformatics, and ecological analysis to gain data from a recently evolved asexual organism. Their study system is quite interesting, as most analyses of asexuals focus on the ancient evolutionary scandals and even the ones dealing with more recently evolved parthenogens are far from the 25 or so years of species age assumed for the crayfish. Thus, the system is unique in allowing to address questions about incipient parthenogenesis. The manuscript clearly is an extension of the original genome paper (Gutekunst et al. 2018), which already showed some population genetic inferences. The authors' data are certainly valuable and need to be published.

There are, however, some shortcomings to the current manuscript:

In summary the manuscript is rather descriptive. Itself this is not bad, however in this particular case there is no real scientific question or hypothesis stated in the introduction and then addressed with the data at hand. Consequently the discussion is rather weak, not really discussing the results in the light of previous findings or theories. I think the authors should be able to improve on this in a revision of the manuscript, as their data allow for comparisons to sexual species. Are there for example no data from a closely related sexual crayfish, e.g. *Procambarus fallax* listed with some populations in the *P. virginalis* genome paper (Gutekunst et al. 2018)? Indirect comparisons with distantly related species might of course be regarded as weak. However, given the amount of population genetic data available from sexual species across the animal tree of life a general comparison between this asexual organisms and an average over data from several sexual species should be possible and will be of interest.

As a general comment on the results, in reading the original genome paper by Gutekunst et al. (2018) it appears to me that the reference genome is not phased and the N50 indicates the presence of many broken contigs. I would guess that this makes the mapping of data undertaken here a bit tricky, as heterozygosity might be due to the triploidy, i.e. internal, not external.

Please note that I cannot comment on the suitability of the statistics and measures used in the ecological assay (p11 bottom - p13)

In particular I like to make the following comments:

p2 lines 35, 38: Throughout the text the claims of "first" appear to be a bit abundant. Even if correct, this should be avoided as it doesn't really matter.

p3 l51: This claim about *Adineta vaga* is now disputed and might very well be an assembly artefact

p3 l52: The nematodes might not be the best example to use here. Their genomics are rather weird (1 chromosome) and the manuscripts are not agreeing on the source of heterozygosity. Equally note that the loss of meiosis genes might be due to a general absence of these genes in the genus.

p3 l57: Shouldn't the "trajectories" be leading to a question? Like this the sentence does not say anything.

p3 l84ff: There is no question or hypothesis stated here.

p5 l114-122: This needs to be set in comparison with other species (see above). The way the numbers are given here, they are just numbers, not even a comparison to the *P. virginalis* genome size is given.

p6 l149: Are these SNV neutral?

p7 l176: Again these data would need to be compared to other species, ideally sexually reproducing crayfish.

p8 l203: Personally, I really dislike this sudden invocation of a link to humans and cancer in particular. To me this is attention fishing. I would assume that tumor evolution can be studied in cell lines (of tumors), whereas the crayfish can tell us something about arthropod genomics.

p10 l261: Why the choice of contigs >10kb?

p11 l274: It is not clear to me what is meant by "reference read" here.

Figure 4 legend (B): I am not sure what can be learnt from this and in fact the whole figure 4B.

I would hope that my comments will help the authors to improve their manuscript.

Best regards

Dr. Philipp Schiffer

Reviewer #2:

Remarks to the Author:

Dear Sir

I read with interest the manuscript "Population Genomic analysis of the monoclonal marbled crayfish" by Olena Maiakovska et al. It is an interesting manuscript on population genomics of a marbled crayfish, a parthenogenetic arthropod of recent evolution. This manuscript was written in a correct and clear English, easy to follow and understand. I found the subject of the manuscript quite interesting since study an aquatic parthenogenetic triploid species at population level force the reader to think out of the box from the most common case that is sexual diploid animal organism.

General comment

Unfortunately, the methods used were not the most appropriate for studying populations at genomic level. I encourage the authors to increase the sample size for the analysis and look for the assistance or participation of a population geneticist, especially with expertise on plant genetics/genomics or phylogeography that are more used to study species with different ploidy levels and clonal reproduction with/without sexual reproduction. Otherwise readers with an expertise in genetics will quickly point out the flaws in the genetic and phylogenetic analyses. Therefore, I recommend the editor to reject the manuscript and encourage the authors to improve sampling in number and diversity of locations and reanalyze the data under a population genomics perspective.

Specific comments

a) P1.L1 The title of the manuscript can be misleading. Most part of the manuscript describes a comparative analysis of SNVs found on crayfish individuals in different populations. The term "population genomic analysis" is used as the analysis of populations using genomic data, in other words, population genetics at genome level. I would rather change the title of the manuscript to something like "comparative analysis of genomic variation" or something similar. This change would avoid criticism of readers coming from the field of population genetics/genomics. Population genomic analysis would need a much larger sampling of the individuals to obtain robust estimations and conclusions about the evolutive forces that could describe the extant found genetic variation.

b) p3.L70 Though not the subject of the manuscript, the notation of the "AA'B" triploid for *Procambarus virginalis* baffles me a bit. Marbled crayfish is a new species derived from slough crayfish (*P. fallax*). Since it is a triploid derived from the same species and not from hybridization with other species, it should be an autopolyploid AAA (highly heterozygous though) instead of AA'B. This nomenclature with letters

refers the set of chromosomes of an organism, where A and B indicates different species (as in allopolyploids). Unfortunately, I miss the A' definition in the AA'B nomenclature.

c) P4.L90 The authors mention that there is no selection. That is true if we talk about sexual selection, however there is a special kind of selection instead, the clonal selection as in somatic evolution in cancer.

d) P5.L111 One individual is not a representation of a population. Further number of individuals need to be sampled in order to improve the population representation.

e) P5.L115 There is only a Methods section but no Materials and Methods section in the manuscript.

f) P5.L118. There is no much detail on how the tree of Fig 2A was constructed. Only with the information indicated in the Methods section, it corresponds to a distance tree (NJ) but not to a phylogenetic tree (ML). In addition, it is difficult to rule out sexual reproduction in this population (not found meiosis doesn't mean that it does not exist at all). One possibility to consider in order to avoid the "noise" of recombination (at very low levels) would be to use a molecular marker that is inherited only maternally with no recombination, which is the mitochondria. In this way, it would be more affordable to sequence mitochondria in many more individuals, and at the same time ruling out the possibility to have even low levels of recombination in your analysis. Another problem with the tree is that there only one individual is considered in the analysis as outgroup, the hypothetical foundational individual; including more. Finally, in order to identify monoclonality, this hypothesis should be tested and contrasted with the alternate hypothesis of polyclonal origin, which, as mention before, needs more sampling size in all the known populations. Visual determination of clonality with a distance tree could be produced for different reasons (even artifacts) and not only because the evolutionary process such as monoclonality.

g) P6.L135 Fig. 3A does not show a clear clustering in two sets, there is one Malagasy individual far from the rest of the individuals.

h) P7.L162 There is insufficient time (2017 to 2019) to SNV patterns to differentiate

k) P9.L221 Large phenotypic plasticity instead of "flexible phenotypes"

l) P10.261 Sequences longer “than”

m) P11.L287 More detail about the protocol for tree construction

o) Text and tables in the manuscript and the supplementary documents should be self-explanatory. Figure legends must include references to colors and indicate clearly what they are. Figure 3 and 4 lack of a color legend (text or image). Figure 4b needs a better resolution to identify red regions.

Reviewer #3:

Remarks to the Author:

This paper is interesting because it is studying, through a population genomic approach, the population structure of a recent parthenogen (the marble crayfish *Procambarus virginalis*). Previous studies have shown that this species originated recently, through autopolyploidization, and since has invaded many new habitats (especially in Madagascar) as a parthenogen. It is a widespread invasive clone with an apparent low genetic variability, as has been reported for the androgenetic *Corbicula* lineages.

The link between clonality, invasion and polyploidy is a critical topic addressed here through an interesting model system, *P. virginalis*. Moreover, the authors included interesting results on population size assessment and simulated the population growth, important parameters when dealing with an invasive species.

I have however some major comments and therefore I recommend publication with major revision.

In the attached pdf file a detailed review is provided.

Reviewer #4:

Remarks to the Author:

I was asked to review the capture-recapture and growth model component of this paper as I am not an expert in genetics, but am in capture-recapture.

My first concern was that the methods for converting estimated abundance to density and for extrapolating the capture-recapture results to further derived quantities was not well explained. Perhaps a flow chart would help. However, more importantly, this very small capture-recapture experiment relative to total population size inevitably led to a total abundance estimate that is so imprecise it is not really informative. $N^{\hat{}}=170,000$ and $SE = 103,166$.

The N point estimate is then plugged into the growth model in a manner that did not propagate the uncertainty in N to the growth model estimates. For these reasons, I would not put much trust in these conclusions.

Reviewer #1

1) As a general comment on the results, in reading the original genome paper by Gutekunst et al. (2018) it appears to me that the reference genome is not phased and the N50 indicates the presence of many broken contigs. I would guess that this makes the mapping of data undertaken here a bit tricky, as heterozygosity might be due to the triploidy, i.e. internal, not external.

>> To avoid confounding by "internal" heterozygosity, we filtered all heterozygous positions in the reference sequence. This is described in the Methods section under Single Nucleotide Variant calling and polymorphic site calculation ("heterozygous sites in the reference sample were eliminated to narrow down the calculation of polymorphic sites in non-heterozygous loci").

2) p2 lines 35, 38: Throughout the text the claims of "first" appear to be a bit abundant. Even if correct, this should be avoided as it doesn't really matter.

>> All the claims were removed (abstract and discussion).

3) p3 l51: This claim about *Adineta vaga* is now disputed and might very well be an assembly artefact

>> We have removed the focus on *Adineta* and nematode genomes in this paragraph. Instead, a stronger emphasis is placed on the comparative genome analysis by Jaron et al. (ref. 8).

4) p3 l52: The nematodes might not be the best example to use here. Their genomics are rather weird (1 chromosome) and the manuscripts are not agreeing on the source of heterozygosity. Equally note that the loss of meiosis genes might be due to a general absence of these genes in the genus.

>> We have removed the focus on *Adineta* and nematode genomes in this paragraph. Instead, a stronger emphasis is placed on the comparative genome analysis by Jaron et al. (ref. 8).

5) p3 l57: Shouldn't the "trajectories" be leading to a question? Like this the sentence does not say anything.

>> Rephrased: "Also, a comparative analysis of 24 asexual genomes has revealed a substantial degree of diversity in their features, which probably reflects biological differences among asexual species⁸."

6) p3 l84ff: There is no question or hypothesis stated here.

>> changed to: "..., but their genomic variation has not been analyzed yet."

7) p5 l114-122: This needs to be set in comparison with other species (see above). The way the numbers are given here, they are just numbers, not even a comparison to the *P. virginalis* genome size is given.

>> A comparison was added: "This is an order of magnitude below the frequency of SNVs in clonal daphniids³¹ and likely reflects the young evolutionary age of marbled crayfish".

8) p6 l149: Are these SNV neutral?

>> The numbers refer to total numbers (not restricted to neutral SNVs).

9) p7 l176: Again these data would need to be compared to other species, ideally sexually reproducing crayfish.

>> We have added the following sentence: "the population size and corresponding population density is consistent with data reported for the closely related freshwater crayfish species *P. fallax* in Florida³⁵ⁿ".

10) p8 l203: Personally, I really dislike this sudden invocation of a link to humans and cancer in particular. To me this is attention fishing. I would assume that tumor evolution can be studied in cell lines (of tumors), whereas the crayfish can tell us something about arthropod genomics.

>> We removed the reference to tumor genome evolution.

11) p10 l261: Why the choice of contigs >10kb?

>> The sentence has been rephrased to clarify the issue: "Also, alignments on short contigs (≤ 10 kb) were discarded to avoid loss of time in processing fragmented sequences."

12) p11 l274: It is not clear to me what is meant by "reference read" here.

>> The sentence was clarified: "Only sites that were covered in the reference were included in the analysis."

13) Figure 4 legend (B): I am not sure what can be learnt from this and in fact the whole figure 4B.

>> The figure legend has been clarified. While the heatmap is somewhat redundant with the PCA, it offers an additional perspective on the patterns of genetic variation. Also, Reviewer #3 labels Fig. 4B as "interesting" (see comment referring to lines 152-153)

Reviewer #2 (Remarks to the Author):

1) P1.L1 The title of the manuscript can be misleading. Most part of the manuscript describes a comparative analysis of SNVs found on crayfish individuals in different populations. The term “population genomic analysis” is used as the analysis of populations using genomic data, in other words, population genetics at genome level. I would rather change the title of the manuscript to something like “comparative analysis of genomic variation” or something similar. This change would avoid criticism of readers coming from the field of population genetics/genomics. Population genomic analysis would need a much larger sampling of the individuals to obtain robust estimations and conclusions about the evolutive forces that could describe the extant found genetic variation.

>> We thank the reviewer for this highly constructive comment. The title was changed to "Genomic variation in the monoclonal marbled crayfish (*Procambarus virginalis*)"

2) p3.L70 Though not the subject of the manuscript, the notation of the “AA'B” triploid for *Procambarus virginalis* baffles me a bit. Marbled crayfish is a new species derived from slough crayfish (*P. fallax*). Since it is a triploid derived from the same species and not from hybridization with other species, it should be an autopolyploid AAA (highly heterozygous though) instead of AA'B. This nomenclature with letters refers the set of chromosomes of an organism, where A and B indicates different species (as in allopolyploids). Unfortunately, I miss the A' definition in the AA'B nomenclature

>> We have renamed the notation to "AAB" and clarified it in the introduction: "It is characterized by a triploid AAB genotype, with a duplicated "A" haplotype and a heterozygous "B" haplotype¹⁷. "

3) P4.L90 The authors mention that there is no selection. That is true if we talk about sexual selection, however there is a special kind of selection instead, the clonal selection as in somatic evolution in cancer.

>> This is a misunderstanding, as we do not mention that there is no selection.

4) P5.L111 One individual is not a representation of a population. Further number of individuals need to be sampled in order to improve the population representation.

>> We addressed this point by recasting our WGS data analysis. Our new Fig. 3 now shows that genetic patterns of animals from a single population are sufficiently specific to separate the Lake Reilingen population from other European populations. It should also be noted that

the marbled crayfish genome is rather large (3.5 Gb), thus requiring a lot of cost-intensive sequencing for the generation of high-quality datasets. As currently constituted, our study is already based on the analysis of 2,000 Gb of DNA sequence.

5) P5.L115 There is only a Methods section but no Materials and Methods section in the manuscript.

>> Our sections are consistent with the journal style.

6) P5.L118. There is no much detail on how the tree of Fig 2A was constructed. Only with the information indicated in the Methods section, it corresponds to a distance tree (NJ) but not to a phylogenetic tree (ML). In addition, it is difficult to rule out sexual reproduction in this population (not found meiosis doesn't mean that it does not exist at all). One possibility to consider in order to avoid the "noise" of recombination (at very low levels) would be to use a molecular marker that is inherited only maternally with no recombination, which is the mitochondria. In this way, it would be more affordable to sequence mitochondria in many more individuals, and at the same time ruling out the possibility to have even low levels of recombination in your analysis. Another problem with the tree is that there only one individual is considered in the analysis as outgroup, the hypothetical foundational individual. Finally, in order to identify monoclonality, this hypothesis should be tested and contrasted with the alternate hypothesis of polyclonal origin, which, as mention before, needs more sampling size in all the known populations. Visual determination of clonality with a distance tree could be produced for different reasons (even artifacts) and not only because the evolutionary process such as monoclonality.

>> Details were clarified in the figure legend and in the Methods section. We now also include a mitochondrial tree, with the *P. fallax* mitochondrial genome sequence as an outgroup (Fig. S1), which further confirms *P. virginalis* clonality.

7) P6.L135 Fig. 3A does not show a clear clustering in two sets, there is one Malagasy individual far from the rest of the individuals.

>> Clustering has been improved by the refinement of the data analysis and the inclusion of an additional dataset. See our revised Fig. 4B.

8) P7.L162 There is insufficient time (2017 to 2019) to SNV patterns to differentiate.

>> We agree with the reviewer. The data is now an important part of our re-cast Fig. 3, where it is interpreted to reflect little genomic variation among animals from a specific local population.

9) P9.L221 Large phenotypic plasticity instead of “flexible phenotypes”

>> Changed as suggested.

10) P10.261 Sequences longer “than”

>> This sentence was rephrased/clarified (see Reviewer #1, point 11).

11) P11.L287 More detail about the protocol for tree construction

>> Complete details (references and parameter settings) are now provided.

12) Text and tables in the manuscript and the supplementary documents should be self-explanatory. Figure legends must include references to colors and indicate clearly what they are. Figure 3 and 4 lack of a color legend (text or image). Figure 4b needs a better resolution to identify red regions.

>> Text, tables and figure legends were revised as suggested by the reviewer. Figure 4B was replaced with an improved version (now Fig. 4C).

Reviewer #3 (Remarks to the Author):

This paper is interesting because it is studying, through a population genomic approach, the population structure of a recent parthenogen (the marble crayfish *Procambarus virginalis*). Previous studies have shown that this species originated recently, through autopolyploidization, and since has invaded many new habitats (especially in Madagascar) as a parthenogen. It is a widespread invasive clone with an apparent low genetic variability, as has been reported for the androgenetic *Corbicula* lineages.

The link between clonality, invasion and polyploidy is a critical topic addressed here through an interesting model system, *P. virginalis*. Moreover, the authors included interesting results on population size assessment and simulated the population growth, important parameters when dealing with an invasive species.

I have however some major comments and therefore I recommend publication with major revision.

This paper does not refer to the manuscript of Kato et al 2016 where the cytological study of oogenesis has been done on *P. virginalis*. This article shows a modified meiosis: the first meiosis-like division being non-reductional with an apparent pairing of chromosomes along

the equatorial plate (characteristic of meiosis I), the second meiosis-like division being equational. Since meiosis is observed in *P. virginalis*, recombination is possible and signatures of recombination should be investigated.

A comparison with *D. pulex* (well-studied at genetic and cytological level): hybridization in *Daphnia* results in a suppression of a normal division in meiosis I (which is the genetic equivalent of a central fusion). As described in Hiruta et al. 2010 (Chromosome res): "In the first meiosis of *Daphnia pulex*, bivalents align at the equatorial plate and begin to separate into two half-bivalents. However, the division is arrested at early anaphase. Then, each half-bivalent moves back and sister chromatids rearrange as a diploid equatorial plate around the equator of the spindle. Finally, the second meiosis-like division takes place normally, producing a single polar body and a daughter cell." The authors describe this mode of parthenogenesis in *D. pulex* as the fusion of meiotic products, although both of the fused products are intermediate because two "complete" haploid nuclei were not formed in the process. Thus, the mode of parthenogenetic oogenesis in *D. pulex* can be thought of as an intermediary mode between "apomixis," representing the remnants of meiosis, and "automixis," resulting in the fusion of meiotic products. I think something very similar is observed by Kato et al for *P. virginalis* with some differences concerning the triploid set of chromosomes. The authors should check all this literature and be more precise in the description of the mode of reproduction of *P. virginalis* in their manuscript without concluding that it is apomictic without recombination.

>> We have rewritten and expanded the text in the introduction (third paragraph) to clarify the issue and also analyzed our datasets for evidence of recombination and polyclonality (see our new Figures 2C and S1). In agreement with the available literature, our findings provide further support for the notion that the mode of parthenogenetic oogenesis is apomixis. Please also see our responses to specific comments below.

I wonder why the authors did not use a nuclear marker instead of a mitochondrial marker to confirm the taxonomic identity of each population since *P. virginalis* is known to have an identical mitochondrial genome to *P. fallax* (where it originated from). How can the authors confirm that the populations (Table S1) tested by PCR are indeed *P. virginalis*?

>> We used both a mitochondrial (cytochrome B) and a nuclear (*Dnmt1*) marker and have clarified this both in the Methods section and in the legend to Tab. S1. We apologize for this oversight.

The authors did not fully exploit their NGS dataset or could have done a more exhaustive population genomic study to eventually detect signatures of recombination. The authors could also study in the different individuals of the different populations whether there is a loss

of heterozygosity (LOH) along the contigs, suggesting eventually some parthenogenetic mode of reproduction with recombination. The crayfish *P. virginalis* is considered clonal based on 6 microsatellite fragments that remained stable over one generation (<https://link.springer.com/article/10.1007/s00114-007-0260-0>). These results do not exclude recombination in some regions not investigated. In automixis with central fusion (or with no separation of homologous chromosomes as observed by Kato et al. 2016), only 50% of the recombination events lead to actual LOH (<https://www.genetics.org/content/206/2/993>) and recombination may be rare. Nevertheless, to check for signatures of recombination a LOH analysis could be done even if LOH between alleles are more difficult to detect because of triploidy. Moreover, some phased segments (haplotypes A, A' and B) of contigs in the ancestral reference genome are needed to compare with the derived samples: check in the different tested individuals if some changes occurred towards AAB, AAA, ABB or A'A'B, etc. Detecting LOH could confirm that haplotypes recombine together (if they recombine at all). If B is too divergent from A and A' (on line 70 the authors should provide the level of heterozygosity measured between A A' and B) it might not recombine and we should not find genotype evolution like AA'B → ABB or BBB or A'BB. To summarize, doing a LOH analysis on the dataset (and ideally having more individuals per population) could provide some indications of recombination going on in *P. virginalis*. It is therefore surprising that the authors removed all heterozygous sites of the reference genome in their comparative analysis which excludes any LOH detection. The only thing the authors considered is a gain of heterozygosity in the studied asexual populations (a gain of heterozygosity is expected to occur through mutations) but not a loss of heterozygosity. I therefore think some interesting results are not looked at here.

>> It should be noted that the evidence for clonality in *P. virginalis* is not limited to the aforementioned microsatellite analysis, but also includes additional observations, which is now explained in more detail in our revised introduction (third paragraph). In agreement with this notion, a comparative analysis of mitochondrial genome sequences also showed clonality (see our new Fig. S1). Furthermore, we performed an LOH analysis on the dataset and did not provide any evidence for recombination (see our new Fig. 2C).

Specific comments on the manuscript:

Introduction

Lines 53-54: In the introduction the authors mention “parthenogenetic genomes evolve towards effective haploidy, as predicted by the Meselson effect”.

I think this is an overstatement since very few genomes have been sequenced from parthenogens and even fewer show such pattern of “effective haploidy”. Moreover, there is

nowadays no chromosome-scale genome assembly of a parthenogenetic species (using long reads and Hi-C) to confirm such haploidy evolution.

>> The corresponding part of the introduction has been rephrased. Also see Reviewer #1, points 3-5.

Line 55: "these processes require asexual reproduction over millions of years" Asexual reproduction is very vast, it includes vegetative reproduction (from somatic cells) as well as parthenogenetic reproduction (from unfertilized gametes). Moreover, within parthenogenesis the authors should distinguish between the different types of parthenogenetic modes including meiosis. If meiosis occurs it may impact the genome structure and heterozygosity and prevent the "Meselson effect", even after millions of years. The authors should be precise in using the terminology of asexuality and parthenogenesis

>> The corresponding part of the introduction has been edited.

Line 67: apomictic parthenogenesis. □ Having read all the manuscripts on marble crayfish and as described above as major comment, the paper of Kato et al. 2016 should be mentioned here and as for *D. pulex* I am not sure how to define this mode of reproduction in *P. virginalis*. Apomictic parthenogenesis could be correct since no fusion of meiotic products occur. However, this is distinct from mitotic cell division because a polar body, a distinguishing feature of meiosis, is emitted at the end of the maturation division. Moreover, recombination could be possible during chromosome pairing at the equatorial plate and it is therefore not sure and demonstrated yet in *P. virginalis* that all offspring are genetically identical. □

>> The introduction has been expanded to explain this point and the paper by Kato et al. has been cited (3rd paragraph).

Line 70: 0,5% average genome heterozygosity.

This is not such a high level of heterozygosity, in many other parthenogens it is higher than this. The level observed in the marble crayfish is higher than the sexual relatives. The authors should be more precise here in what is meant with "high".

>> We agree with the reviewer and have deleted the reference to "high" genome heterozygosity. Also, our most recent analyses suggests that we have previously underestimated the level of genome heterozygosity in the sexually reproducing mother species (*P. fallax*) and that the difference between marbled crayfish and *P. fallax* is not as high as originally thought.

Line 90: the absence of recombination in the crayfish remains to be shown!

>> We have rephrased the sentence: "Similarly, if recombination is absent, ...".

Results

Line 99: "other information sources" Which other information sources?

>> Clarified: "unpublished information from the scientific community"

Figure 1: in the legend the authors should refer to Supp Table 1 (if reader wants to know the exact location).

>> Done as suggested.

Table S1. □The authors should include a column here indicating how many individuals were tested per population (Line 103 the authors mention: two or more animals).

>> The number of tested animals is now indicated in Table S1.

Line 118: "ranged from 5722 to 6007". □However, in Fig 2C (line 580) the number of SNVs do not match with this number around 6000. It appears to be around 1500 in Fig 2C. Where is the rest? Is it from other filters/data? □

>> The entire SNV analysis was revised and refined. New numbers were calculated and provided in figures 3 and 4 and in the Results section.

Lines 124-125: the PCA (Figure 2B) The PCA is not informative and both axes only account for 12,6% and 12% of the variability. I would remove this analysis from the manuscript.

>> The PCA (now Fig. 3B) has been revised and shows that genetic patterns of multiple animals from a single population (Lake Reilingen) are sufficiently specific to separate the population from other European populations.

Lines 133: SNVs between Europe and Madagascar. Can the authors also define here the number of unique SNVs for Europe and for Madagascar and provide a figure as in Fig 2C. Again, the PCA is not very informative (PC1 and PC2 axes low) and could be removed, the Venn diagram is clear.

>> The entire SNV analysis was revised and refined. New Venn diagrams and PCAs were provided in figures 3 and 4.

Lines 140-142: this sentence is not clear (the authors did not compare the population from Madagascar with the ancestral aquarium one)

>> This sentence has been clarified: "Furthermore, the considerable number of shared polymorphisms between European and Malagasy animals (Fig. 4A) likely reflects the

distance of the wild populations from the ancestral aquarium lineage, which also provided the reference genome sequence{Gutekunst, 2018 #2396}.".

Lines 152-153: again the PCA is not informative (Fig 4A). The heatmap (Fig 4B) is interesting but the ordered clusters are not visible (the lines delimiting the clusters should be more stretched vertically). Moreover, the authors should indicate for population Lake Reilingen which individual is from 2017 and which ones from 2019.

>> Our revised PCA (now Fig. 4B) shows a clear separation of European and Malagasy populations. We also revised the heatmap (now Fig. 4C) according to the reviewer's suggestions.

Table S5: could the authors add here the number of individuals sampled daily (the data should be available to reconstruct Fig S1)

>> Fig. S1 (now Fig. S3A) shows animal numbers after application of the Schnabel formula. As such, the Figure cannot be directly reconstructed from the data in Tab. S5. This is now clarified in the figure and in the figure legend.

Lines 181-182: the SNV dataset of 4 specimens is poor to estimate the total genetic variability of the population, not sure if this is sufficient for this growth model

>> The genetic variability of the 5 specimens from Lake Reilingen (Fig. 3) showed a consistently low level of genetic variability that way also comparable to the complete dataset. As such, we felt that it was sufficient for a first growth model. As the results are overall consistent with empirical data about marbled crayfish population growth, we consider the growth model a valuable addition to our paper.

Fig 5D (Lines 190-191): I do not see how the figure shows this high convergence mentioned between genetic variability and the simulated exponential growth at 3.7 years, both the Allee growth and the exponential growth has an intersection with the dashed horizontal line.

>> The sentence was rephrased to clarify its meaning: "Further analysis showed that the genetic variability with a simulated exponential growth intersect with the variability of the current Lake Reilingen population after 3.7 (± 0.3) years (Fig. 4D).".

Discussion

Lines 219: a general-purpose genotype has been proposed for the widespread asexual darwinulid ostracod species *Darwinula stevensoni* (check the reference Van Doninck et al. 2002 and 2003). □The discussion is only referring to the young age to explain the low genetic variation, I agree this is one of the main reasons. However, the different modes of

parthenogenetic reproduction also influence the genetic variability. The authors should discuss this.

>> The topic has been added to the discussion and the papers have been cited.

Reviewer #4

My first concern was that the methods for converting estimated abundance to density and for extrapolating the capture-recapture results to further derived quantities was not well explained. Perhaps a flow chart would help. However, more importantly, this very small capture-recapture experiment relative to total population size inevitably led to a total abundance estimate that is so imprecise it is not really informative. $N^{\wedge}\{hat\}=170,000$ and $SE = 103,166$.

>> We have thoroughly revised the corresponding section of the manuscript: (1) We experimentally determined the trapping size cutoff for the used traps (>6 cm, Fig. S2). (2) We analyzed the relationship between the proportion of marked animals and the total of marked animals. A linear regression (Fig. S3A) had an adjusted R-square = 0.9171 ($p=8.312e-06$), indicating that the Schnabel model is valid. (3) We applied a non-linear least square (NLS) regression to predict the catch size plateau (Fig. S3B). This approach is now clearly explained in the text (Results and Methods sections) and illustrated by 2 new supplementary figures (Fig. S2 and S3).

Importantly, our revised approach also allowed us to obtain a much more accurate population size estimate of 22,962 ($SE=3,613$) sexually mature animals >6 cm. For the total population, the revised size estimate is 192,000 ($SE=67,000$) animals. We explain in the text that the high error of this estimate is largely due to the difficulty to accurately determine the number of small animals. Nevertheless, the population size and corresponding population density is consistent with data reported for the closely related freshwater crayfish species *P. fallax* in Florida (ref. 35).

The N point estimate is then plugged into the growth model in a manner that did not propagate the uncertainty in N to the growth model estimates. For these reasons, I would not put much trust in these conclusions.

>> We introduced uncertainties about the growth rate and the population size estimate into our growth models. Our revised Fig. 5D includes corresponding windows of uncertainty.

Reviewers' Comments:

Reviewer #1:

Remarks to the Author:

Dear editor,

The authors have sufficiently addressed all issues I have raised in the first review of their manuscript. At this point I have only minor comments:

line 64 on page 3: is there a reference missing here?

lines 214 to 216 on page 9: I do not understand the sentence starting with "We have enabled..." line

lines 233 to 235 on page 9: I think the reference to methylation comes a bit out of the blue and without further data should not be mentioned even. The reference to "non-genetic regulatory mechanisms of adaptability" is misleading, as methylation is a genetic mechanism.

Table S1 should contain the SRA accession numbers for the WGS datasets and also Genbank accessions for the sequences obtained by PCR.

Kind regards,

Dr. Philipp Schiffer

Reviewer #2:

Remarks to the Author:

General comments:

I read with interest the rebuttal letter to the reviewers of the manuscript "Genomic variation in the monochlorinated marbled crayfish" of Maiakovska et al. I feel that with the inclusion of the new text and data and the pertinent answers to the questions raised by the reviewers this manuscript is complete and more useful to the readers of this manuscript. Therefore, I recommend the publication of this manuscript with minor revision from my side.

Specific comments:

P3 L64: missing reference (ref.)

P4 L95: "classical attributes of populations" should be referred as "classical attributes of panmictic populations"

P4 L97-99: substitution of "and by the observation of" by "due to" or "produced by" may give more clarity to the phrase since the changes in allele frequencies are the observed effects of processes such as genetic drift, gene flow and natural selection in sexually reproducing organisms. In any case this statement needs to be rephrased since there will be mutation, creation of new alleles and selection in the clonal population. Only genetic drift and gene flow are (typically) present in sexually reproducing organisms.

P6 L130: Please include in the text or in supplementary data how the mitochondrial tree was constructed and if the data was coming from specific sequencing of mitochondria, assembly or mapping reads against a reference mitochondrial genome. If assembled, please make the data available in a public database.

P6 L155: please reference the Malagasy samples data if it is not original data and/or included in the

supplementary data as an additional table.

P11 L289: the more suitable name for the section would be "Tree construction". Phylogenetic trees are made from sequence data using a robust method such as maximum likelihood (ML). The tree presented in Fig 2A is a distance tree using a clustering method (NJ).

P27 Fig 3B. Including the Heidelberg individual in the PCA analysis make the clustering to be observed in "one dimension", in order to see with the PCA how similar they are in two dimensions it would be better not to include this individual outlier that reduces the power of the PCA.

Table S2: correct "Reilinger" with "Reilingen". In addition, to compare tables S1 and S2, there should be an extra column in table S2 indicating the corresponding populations from table S1 and sort table S2 accordingly.

Reviewer #3:

Remarks to the Author:

The authors have revised their manuscript and taken into account most comments of the reviewers. I suggest the manuscript is published in Communications biology but I still suggest some minor modifications.

Reviewer #4:

Remarks to the Author:

The authors have satisfactorily addressed my concerns about the mark recapture and population simulation portions of this manuscript about which I was asked to comment.

Reviewer #1:

1. line 64 on page 3: is there a reference missing here?

>> The reference has been added.

2. lines 214 to 216 on page 9: I do not understand the sentence starting with "We have enabled..."

>> The sentence has been clarified: "We provide an approach for the analysis of clonal population patterns in the absence of allele frequency dynamics by using whole genome sequencing for the detection of rare SNVs."

3. lines 233 to 235 on page 9: I think the reference to methylation comes a bit out of the blue and without further data should not be mentioned even. The reference to "non-genetic regulatory mechanisms of adaptability" is misleading, as methylation is a genetic mechanism.

>> We have rephrased the corresponding sentences and removed the reference to methylation: "The analysis of marbled crayfish genomes will facilitate our understanding of the mechanisms of asexual genome diversification and adaptation. This could conceivably include the preservation of high heterozygosity levels, the selection of specific genetic variants and changes in genome regulatory mechanisms^{43,44}."

4. Table S1 should contain the SRA accession numbers for the WGS datasets and also Genbank accessions for the sequences obtained by PCR.

>> Accession numbers for the WGS and PCR amplicon sequencing datasets are provided in the Data Availability part of the Methods section and the corresponding NCBI BioProject has been published (PRJNA599283).

Reviewer #2:

1. P3 L64: missing reference (ref.)

>> The reference has been added.

2. P4 L95: "classical attributes of populations" should be refereed as "classical attributes of panmictic populations"

>> Corrected as suggested.

3. P4 L97-99: substitution of "and by the observation of" by "due to" or "produced by" may give more clarity to the phrase since the changes in allele frequencies are the observed effects of processes such as genetic drift, gene flow and natural selection in sexually reproducing organisms. In any case this statement needs to be rephrased since there will be mutation, creation of new alleles and selection in the clonal population. Only genetic drift and gene flow are (typically) present in sexually reproducing organisms.

>> Corrected as suggested: "Hence, the population genetic structure of clonal marbled crayfish cannot be characterized by allele frequency dynamics, as the underlying processes are typical for sexually reproducing populations^{30,31}."

4. P6 L130: Please include in the text or in supplementary data how the mitochondrial tree was constructed and if the data was coming from specific sequencing of mitochondria,

assembly or mapping reads against a reference mitochondrial genome. If assembled, please make the data available in a public database.

>> This is now explained in the Methods section (chapter “Tree construction”).

5. P6 L155: please reference the Malagasy samples data if it is not original data and/or included in the supplementary data as an additional table.

>> The Malagasy sample is now referenced in Table S2.

6. P11 L289: the more suitable name for the section would be “Tree construction”. Phylogenetic trees are made from sequence data using a robust method such as maximum likelihood (ML). The tree presented in Fig 2A is a distance tree using a clustering method (NJ).

>> Corrected as suggested.

7. P27 Fig 3B. Including the Heidelberg individual in the PCA analysis make the clustering to be observed in “one dimension”, in order to see with the PCA how similar they are in two dimensions it would be better not to include this individual outlier that reduces the power of the PCA.

>> Corrected as suggested.

8. Table S2: correct “Reilinger” with “Reilingen”. In addition, to compare tables S1 and S2, there should be an extra column in table S2 indicating the corresponding populations from table S1 and sort table S2 accordingly.

>> Corrected as suggested.

Reviewer #3:

1. Lines 65-66: automixis is not only characterized by the fusion of meiotic products, in most species it is characterized by the absence of separation of homologous chromosomes during meiosis one (all homologous chromosomes pair, eventually recombination takes place but then there is no cell division and all homologous chromosomes are retained. Apomixis is normally bypassing meiosis. The authors should be clear here when defining automixis and apomixis.

>> Clarified to emphasize that automixis generates genetically distinctive offspring, while apomixis generates clonal offspring: “Automixis is characterized by the fusion of meiotic products, with the production of genetically distinctive offspring due to genetic recombination. Apomixis, on the other hand, is characterized by a bypass of meiotic recombination, resulting in clonal progeny that represent an exact copy of the maternal genotype.”

2. Lines 72-75: I do not agree with the description of the reproductive mode here by the authors. If meiosis is observed (but defective) it is a modified meiosis that is taking place in this parthenogenetic mode of reproduction and not a simple mitotic division. Meiosis and mitosis also differ strongly in the molecular actors involved. If this would be studied in the marbled crayfish I bet it would be actors involved in the meiotic process. Therefore, the authors can also not conclude it is apomixis. The authors should simply mention what is observed in Kato et al.

>> Corrected as suggested.

3. Lines 78-79: as suggested by reviewer #2 I agree the nomenclature AAB is confusing for an autopolyploid species. Rather use AAA'. The sentence: "It is characterized by a triploid AAB genotype, with a duplicated "A" haplotype and a heterozygous "B" haplotype might still be confusing to the reader, the authors should specify the autopolyploidy here.

>> During the previous revision, we have changed the notation to "AAB", as suggested by Reviewer #2 (original point 2). We have now further clarified the sentence: "It is characterized by a triploid AAB genotype, with a duplicated "A" haplotype resulting from autopolyploidization and a heterozygous "B" haplotype, presumably resulting from fertilization¹⁸."

4. Lines 130-134: what is the genetic divergence between these European populations of Fig. 2A? Can a table of genetic divergence be given? Sharing the same mitochondrial genome does not confirm clonality. What is the result of the nuclear gene Dnmt1? Can a similar figure as Fig.S1 be given?

>> The genetic divergence is indicated by the scale bar in Fig. 2A. One "count" represents one SNV from the common set of SNVs for all 11 samples. This is now clarified in the figure legend.

A Dnmt1 tree would require the re-mapping of all WGS datasets. We regret that this is not feasible in the timeframe of this revision.

5. Lines 141-142: absence of LOH suggests no recombination (it could still be automictic parthenogenesis). In the discussion the authors should also be more cautious (lines 210-211).

>> Rephrased to clarify that the results are consistent with the prevailing view: "... consistent with the notion that marbled crayfish reproduce by apomictic parthenogenesis^{14,15}."

6. General comment on the M&M: As also pointed out by reviewer #1, is the reference genome phased? How did the authors eliminate all heterozygous sites in the reference sample and why, this reduces the number of informative regions?

>> The reference genome is not phased, see ref. 18. The elimination of heterozygous sites was necessary to streamline the analysis in the triploid background. The approach is now explained in the Methods section: "To streamline the analysis in the triploid background, heterozygous sites in the reference were eliminated, which focused the analysis of polymorphic sites on non-heterozygous loci. Polymorphic sites were identified by the presence of a single nucleotide substitution relative to the reference sequence. Other types of substitutions such as insertions or deletions were also discarded from the analysis."

7. Lines 296-300 for the study on the loss of heterozygosity: "a minimum number of reference observations of 6", what do the authors mean here? In the analysis, consecutive sites where LOH is observed (in each sample) should be tracked and in Fig. 2C the samples should be compared to the reference genome (and not to each other).

>> The sentence was rephrased to: "Sites were filtered for a Phred-scaled quality score >30 and a sequencing coverage >5X". Fig. 2C was revised to include a highlighted (colored) line representing the reference genome.